# mRNA-Loaded Lipid Nanoparticles Targeting Dendritic Cells for Cancer Immunotherapy

**DOI:** 10.3390/pharmaceutics14081572

**Published:** 2022-07-28

**Authors:** Kosuke Sasaki, Yusuke Sato, Kento Okuda, Kazuki Iwakawa, Hideyoshi Harashima

**Affiliations:** Laboratory for Molecular Design of Pharmaceutics, Faculty of Pharmaceutical Sciences, Hokkaido University, Kita-12, Nishi-6, Kita-Ku, Sapporo 060-0812, Japan; kskssk.sw.77@gmail.com (K.S.); okuken@eis.hokudai.ac.jp (K.O.); kazukiiwak78@eis.hokudai.ac.jp (K.I.)

**Keywords:** lipid nanoparticles, particle size, dendritic cells, mRNA, delivery, cancer immunotherapy

## Abstract

Dendritic cells (DCs) are attractive antigen-presenting cells to be targeted for vaccinations. However, the systemic delivery of mRNA to DCs is hampered by technical challenges. We recently reported that it is possible to regulate the size of RNA-loaded lipid nanoparticles (LNPs) to over 200 nm with the addition of salt during their formation when a microfluidic device is used and that larger LNPs delivered RNA more efficiently and in greater numbers to splenic DCs compared to the smaller counterparts. In this study, we report on the in vivo optimization of mRNA-loaded LNPs for use in vaccines. The screening included a wide range of methods for controlling particle size in addition to the selection of an appropriate lipid type and its composition. The results showed a clear correlation between particle size, uptake and gene expression activity in splenic DCs and indicated that a size range from 200 to 500 nm is appropriate for use in targeting splenic DCs. It was also found that it was difficult to predict the transgene expression activity and the potency of mRNA vaccines in splenic DCs using the whole spleen. A-11-LNP, which was found to be the optimal formulation, induced better transgene expression activity and maturation in DCs and induced clear therapeutic antitumor effects in an E.G7-OVA tumor model compared to two clinically relevant LNP formulations.

## 1. Introduction

Nucleic acid vaccines are being evaluated for use in a number of clinical applications, including cancer, allergy, and infectious diseases [1,2,3,4,5]. It is particularly noteworthy that mRNA was first commercialized as a vaccine against the coronavirus disease 2019 (COVID-19) for the treatment of infections caused by the severe acute respiratory syndrome coronavirus 2 (SARS-CoV-2) [6,7,8], and its application to a wide variety of diseases is now being investigated. The mRNA molecule, unlike DNA, has no risk of unintentional insertion into genomic DNA [9]. In addition, mRNA does not require nuclear delivery and can efficiently express the encoded gene(s) if it is properly introduced into the cytoplasm. Recent developments in mRNA engineering, such as capping technology, sequence design technology, and chemically modified bases, as well as relatively simple synthesis by in vitro transcription, have greatly contributed to the development of this technology [10]. In addition, mRNA can express any peptide and protein structures depending on the sequence design, making it possible to present a variety of antigen peptides to major histocompatibility complex (MHC) classes I and II [11]. In addition, since mRNAs exhibit immunostimulatory properties through recognition by pattern recognition receptors (PPRs), including Toll-like receptors (TLRs) and retinoic acid-inducible gene I (RIG-I)-like receptor (RLR) families [12], which recognize pathogen-associated molecular patterns (PAMPs), they can activate innate immunity efficiently without the need for additional adjuvants [2,13].

While the above-mentioned advantages of mRNA and the progress in mRNA engineering technologies are encouraging the realization of its clinical applications, properties of the mRNA, including instability in an in vivo environment and cell membrane impermeability, prevent it from reaching the cytoplasm efficiently, where it functions. Therefore, it is essential to develop a technology to efficiently deliver mRNA to the cytoplasm of target cells. mRNA delivery systems include lipoplex and lipid nanoparticles (LNPs) [2,14,15,16,17,18]. A Comirnaty (Pfizer (New York, NY, USA)/BionTech (Mainz, Germany)) and Spikevax (Moderna (Cambrige, MA, USA)) are examples of LNP-based mRNA vaccines against COVID-19 [8].

LNPs are composed of pH-sensitive cationic lipids (CLs), phospholipids, cholesterol (Chol), and polyethylene glycol (PEG)-modified lipids. Chol contributes to the stability of LNPs [19,20]. Phospholipids mainly contribute to the formation and stabilization of the lipid membrane [21]. PEG lipids contribute to the regulation of particle size, dispersion stability, and blood retention [22,23,24]. pH-sensitive cationic lipids contain tertiary amino group(s) and are positively charged in a weakly acidic environment [25,26]. This is important for efficient mRNA encapsulation through electrostatic interactions. It is also important for delivering mRNA to the cytoplasm by escape via membrane fusion from acidified endosomes after the LNPs have been internalized into cells through endocytosis [27,28]. Therefore, the lipid composition of the LNPs has significant effects on physicochemical properties as well as potency for drug delivery and, therefore, should be carefully optimized for each specific application and payload due to differences in the properties needed for the specific targeting of cell types or applications [29,30,31].

Dendritic cells (DCs) are able to initiate antigen-specific immune responses in lymphoid tissues after infectious pathogens are sensed and are among the most powerful antigen-presenting cells (APCs) [32,33]. Therefore, DCs are attractive APCs to be targeted for vaccination. However, the systemic delivery of mRNA to DCs faces a number of technical challenges. Although an ex vivo pulsed DC vaccine would have great potential because of its specificity, flexibility, and efficiency [34,35], it requires a complicated process that involves collecting a sample from the patient to re-administer to the patient. The direct targeting of DCs in vivo would be a good strategy for overcoming that drawback. Several studies have reported on DC targeting by modifying ligand molecules on the surface of the nanoparticles [36,37]. While this approach is widely used to achieve specific targeting, the formulation process is very complex and often poses challenges in terms of cost, reproducibility, and difficulty in characterization. On the other hand, Kranz, et al. reported on an RNA-lipoplex (LPX) that was prepared by mixing mRNA and ligand-unmodified liposomes composed of 1,2-di-*O*-octadecenyl-3-trimethylammonium propane (DOTMA) and 1,2-dioleoyl-sn-glycero-3-phosphoethanolamine (DOPE) [2]. Focusing on charge balance, specific gene expression in the spleen was achieved by designing a formulation with a negative charge ratio, which resulted in inducing a relatively selective gene expression in DCs. Antigen-specific antitumor immunity was induced in several cancer models by using an mRNA encoding a cancer antigen. Currently, BioNTech is conducting phase II clinical trials on an mRNA cancer vaccine formulation containing a fixed set of cancer-associated antigens (referred to as the FixVac platform) against advanced melanomas (ClinicalTrials.gov Identifier: NCT04526899) and human papillomavirus 16-positive head and neck squamous cell carcinomas (AHEAD-MERIT study, ClinicalTrials.gov Identifier: NCT04534205) as of the writing this manuscript. However, there are still few reports of LNPs that enable efficient mRNA delivery to DCs after systemic administration, and there is clearly substantial room for improvement regarding efficiency. In this study, we optimized mRNA-loaded LNPs targeting DCs using our originally developed CLs and the recently reported broad particle size control technology of LNPs synthesized using a microfluidic device [38,39] and compared the optimized LNPs with clinically-relevant LNP formulations.

## 2. Materials and Methods

### 2.1. Materials

The pH-sensitive cationic lipid, 7-(4-(dipropylamino)butyl)-7-hydroxytridecane-1,13-diyl dioleate (CL4H6), was synthesized as described previously [38]. 7-Hydroxy-7-(4-morpholinobutyl)tridecane-1,13-diyl dioleate (CL7H6) was synthesized as described in the Supplementary Method. Chol was purchased from SIGMA Aldrich (St. Louis, MO, USA). DOTMA, 1,2-distearoyl-*sn*-glycero-3-phosphocholine (DSPC), DOPE, 1,2-dimirystoyl-*rac*-glycero, methoxyethyleneglycol 2000 ether (PEG-DMG), and 1,2-distearoyl-*rac*-glycero, methoxyethyleneglycol 2000 ether (PEG-DSG) were obtained from the NOF Corporation (Tokyo, Japan). DLin-MC3-DMA was purchased from Selleck Biotech (Houston, TX, USA). 3,3′-Dioctadecyloxacarbocyanine perchlorate (DiO) and Ribogreen were purchased from ThermoFisher Scientific (Waltham, MA, USA). Purified anti-mouse CD16/32 antibody (clone 93), PE Anti-mouse CD40 antibody (clone FGK45), FITC anti-mouse CD80 antibody (clone 16-10A1), phycoerythrin (PE) anti-mouse CD86 antibody (clone A17199A), allophycocyanin (APC) anti-mouse CD69 antibody (clone H1.2F3), PE anti-mouse CD3 antibody (clone 17A2), peridinin-chlorophyll-protein (PerCP)/Cyanine 7 anti-mouse CD19 antibody (clone 6D5), FITC anti-mouse NK1.1 antibody (clone PK136), APC anti-mouse CD11c antibody (clone N418), PerCP/Cyanine 5.5 anti-mouse I-A/I-E antibody (clone M5/114.15.2), and propidium iodide (PI) were purchased from BioLegend (San Diego, CA, USA). Anti-human polo-like kinase 1 (hPLK1) siRNA (siPLK1, sense: 5′-AGA uCA CCC uCC UUA AAu AUU-3′; antisense; 5′-UAU UUA AGG AGG GUG AuC UUU-3′, 2′-OMe-modified nucleotides are in lower case) was purchased from Hokkaido System Science Co., Ltd. (Sapporo, Japan). The ovalbumin (OVA)-encoding mRNA (cat# L-7610) and enhanced green fluorescent protein (EGFP)-encoding mRNA (cat# L-7601) were purchased from Trilink BioTechnologies (San Diego, CA, USA). A microfluidic device, an invasive lipid nanoparticle production (iLiNP) device, was fabricated as described previously [40].

### 2.2. Mice and Cell Cultures

E.G7-OVA cells, generated by transducing the chicken OVA gene into the murine lymphoma cell line EL4, were obtained from the American Type Culture Collection (Manassas, VA, USA). E.G7-OVA cells were grown in Roswell Park Memorial Institute (RPMI) 1640 (Sigma Aldrich, St. Louis, MO, USA) supplemented with 50 μM of β-mercaptoethanol, 10% fetal calf serum, 10 mM 2-[4-(2-Hydroxyethyl)-1-piperazinyl]ethanesulfonic acid (HEPES), 1 mM sodium pyruvate and 100 units/mL of penicillin/streptomycin. Cells were cultured at 37 °C in a 5% CO_2_ incubator.

C57BL/6N mice (female, 6 weeks of age) were purchased from Japan SLC (Shizuoka, Japan).

### 2.3. In Vitro Transcription of mRNA

Nanoluciferase (Nluc)-encoding mRNA was synthesized from linearized pDNA by in vitro transcription using an mMESSAGE mMACHINE T7 Transcription Kit (ThermoFisher Scientific Inc., Waltham, MA, USA) according to the manufacturer’s protocol. The in vitro transcribed mRNA was purified using a MEGAclear Transcription Clean-Up Kit (ThermoFisher Scientific Inc., Waltham, MA, USA) according to the manufacturer’s protocol. The purified mRNA was qualified by denatured agarose gel electrophoresis, was quantified by absorbance, and was then stored at −80 °C until used.

### 2.4. Preparation of RNA-Loaded LNPs

An ethanol solution containing a pH-sensitive cationic lipid, a phospholipid, chol, and PEG-lipid at the indicated molar ratios was prepared at a total lipid concentration of 16 mM. The RNA was dissolved in 25 mM acetate buffer (pH 4.0) containing NaCl (0 to 400 mM). For MC3-LNPs, a fixed lipid composition of MC3:DSPC:chol:PEG-DMG = 50:10:40:1.5 (molar ratio) and acetate buffer without NaCl were used. LNPs were prepared by mixing the lipids in ethanol and an aqueous solution of mRNA using an iLiNP device at a total flow rate (TFR) of 0.5 mL/min and RNA-to-lipid flow rate ratio (FRR) of 3. RNA-to-lipid ratio was adjusted to 26.6 µg of RNA/µmol total lipid. Syringe pumps (Harvard apparatus, MA, USA or YMC Co., Ltd., Kyoto, Japan) were used to control the flow rate. The resulting LNP solution was then dialyzed for 2 h or more at 4 °C against 20 mM MES buffer (pH 6.0), followed by phosphate-buffered saline without Ca^2+^ and Mg^2+^ (PBS(-)) using Spectra/Por 4 dialysis membranes (molecular weight cut-off 12,000–14,000 Da, Spectrum Laboratories, Rancho Dominguez, CA, USA) to remove ethanol and adjust pH to neutral. For screening A and B, 0.1 mol% of DiO was added to the lipid solution, and a mixture of siPLK1 and mNluc at 19:1 (weight ratio) was used.

### 2.5. Preparation of RNA-Lipoplexes (RNA-LPX)

A lipid film was formed by the evaporation of an ethanolic solution containing DOTMA and DOPE at a molar ratio of 1:1 (10 mM, 400 µL). PBS(-) (800 µL) was then added to the lipid film, followed by incubation for 7 min at room temperature to hydrate the lipids. To form empty liposomes, the lipid film was then sonicated for approximately 30 s in a bath-type sonicator. After dilution with PBS(-) 5.92 times (final lipid concentration of 0.845 mM), equal volumes of the liposome suspension and the mRNA solution (0.2 mg/mL in PBS(-)) were mixed to form RNA-LPX with a nitrogen per phosphate ratio of 1.3/2. The resulting sample was used in in vivo experiments without further purification.

### 2.6. Characterization of RNA-Loaded LNPs

The ζ-average size, polydispersity index (PdI), and ζ-potential of the LNPs were measured by means of a Zetasizer Nano ZS ZEN3600 instrument (Malvern Instruments, Worchestershire, UK). The encapsulation efficiency and total concentration of RNA were measured by a Ribogreen assay, as described previously [41].

### 2.7. In Vivo Screening of RNA-Loaded LNPs

C57BL/6N mice were intravenously injected with the RNA-loaded LNPs at a dose of 1 mg RNA/kg (0.05 mg Nluc mRNA/kg). At twenty-four hours after the injection, the liver and half of the spleen tissues were harvested, frozen in liquid nitrogen, and stored at −80 °C for use in Nluc assays. Splenocytes were dissociated from the remaining spleen tissues and were then passed through a cell strainer (40 µm pore, BD Falcon, CA, USA). The recovered cells were spun down (400× *g*, 4 °C, 5 min) to remove the supernatant, resuspended in red blood cell (RBC) Lysis buffer (1 mL, BioLegend) and incubated for 5 min at room temperature. The resulting treated cells were washed with Hanks’ balanced salt solution (HBSS(-)) by spinning (400× *g*, 4 °C, 5 min). The concentration of cells was adjusted to 1 × 10^7^ cells/mL with fluorescence-activated cell sorting (FACS) buffer, and the resulting cells were treated with a 10 µg/mL solution of an anti-mouse CD16/32 antibody followed by incubation at 4 °C for 10 min. The resulting sample was then treated with fluorophore-conjugated anti-mouse antibodies at 4 °C for 30 min. Cells were washed with FACS buffer twice, filtered via a nylon filter, stained with propidium iodide (PI) (BioLegend), and analyzed for cellular uptake and cell sorting (SONY SH800 cell sorter, SONY, Tokyo, Japan). Three thousand living B cells (defined as PI^−^CD19^+^ cells), macrophages (Mac, defined as PI^−^F4/80 ^+^CD11c^+^ cells), and DCs (defined as PI^−^I-A/I-E^+^CD19^−^ cells) were collected in a sample tube containing 20 µL of 2 × passive lysis buffer.

Nluc activity was measured using the Nano-Glo Luciferase Assay Kit (Promega Corporation, Madison, WI, USA) according to the manufacturer’s protocol. Luminescence was measured using a luminometer (Luminescencer-PSN, ATTO, Tokyo, Japan). For tissues, Nluc activity was corrected for protein quantity using a Pierce BCA Protein Assay Kit (ThermoFisher, Waltham, MA, USA) and expressed as relative light units (RLU) per mg protein. Cellular Nluc activity was expressed as RLU per 3000 cells.

### 2.8. Comparison of Transgene Expression Activity

At the tissue level, C57BL/6N mice were intravenously injected with Nluc mRNA-loaded A-11-LNPs, MC3-LNPs or RNA-LPX at a dose of 0.5 mg mRNA/kg. Twenty-four hours after the injection, liver, spleen, and inguinal lymph node (LN) were harvested, frozen in liquid nitrogen, and stored at −80 °C for use in a Nluc assay, as described in Section 2.7.

For splenic DC level, C57BL/6N mice were intravenously injected with EGFP mRNA-loaded A-11-LNPs, MC3-LNPs or RNA-LPX at a dose of 0.5 mg mRNA/kg. Twenty-four hours after the injection, the spleen was harvested. Splenocytes were dissociated from the remaining spleen tissues and were passed through a cell strainer (40 µm pore). The recovered cells were spun down (400× *g*, 4 °C, 5 min), and the supernatant was discarded. The resulting cells were resuspended in RBC Lysis buffer (1 mL) and incubated for 5 min at room temperature. These treated cells were washed with HBSS(-) by spinning (400× *g*, 4 °C, 5 min). The concentration of cells was adjusted to 1 × 10^7^ cells/mL with FACS buffer, and the resulting cells were treated with a 10 µg/mL solution of an anti-mouse CD16/32 antibody and then incubated at 4 °C for 10 min. The resulting material was then treated with an APC anti-mouse CD11c antibody and a PerCP/Cyanine 5.5 anti-mouse I-A/I-E antibody at 4 °C for 30 min. The resulting cells were washed with FACS buffer twice, filtered via a nylon filter, and stained with PI, and the DCs were then analyzed for EGFP and I-A/I-E expression using CytoFLEX (Beckman Coulter, Inc., Brea, CA, USA).

### 2.9. Measurement of Activation Markers in Splenocytes

C57BL/6N mice were intravenously injected with OVA mRNA-loaded A-11-LNPs, MC3-LNPs or RNA-LPX at a dose of 0.03 mg mRNA/kg. Spleen tissues were harvested, and the resulting dissociated splenocytes were passed through a cell strainer (40 µm pore) 24 h after the injection. The recovered cells were spun down (400× *g*, 4 °C, 5 min), the supernatant discarded, and the resulting cells were resuspended in RBC Lysis buffer (1 mL) and incubated for 5 min at room temperature. These treated cells were washed with HBSS(-) by spinning (400× *g*, 4 °C, 5 min). The concentration of cells was adjusted to 1 × 10^7^ cells/mL with FACS buffer, and the resulting cells were treated with a 10 µg/mL solution of an anti-mouse CD16/32 antibody, followed by incubation at 4 °C for 10 min. This preparation was then treated with fluorophore-conjugated antibodies at 4 °C for 30 min. Cells were washed with FACS buffer twice, filtered via nylon filter, stained with PI, and applied for flowcytometric analysis (CytoFLEX). DCs, T cells, B cells, and NK cells were defined as PI^−^I-A/I-E^+^CD11c^+^ cells, PI^−^CD3^+^CD19^−^ cells, PI^−^CD3^−^CD19^+^ cells, and PI^−^CD3^−^CD19^−^NK1.1^+^ cells, respectively. Relative expression of CD40, CD80, CD86 in DCs and CD69 in B cells, T cells, and NK cells were measured.

### 2.10. Prophylactic and Therapeutic Antitumor Effect of mRNA-Loaded LNPs on Mice

For the prophylactic antitumor experiment, C57BL6/N mice (6 weeks old) were intravenously injected with OVA mRNA-loaded A-11-LNPs at the indicated doses on day −14 and −7. On day 0, E.G7-OVA cells (8 × 10^6^ cells/50 μL/mouse) were inoculated to the immunized mice subcutaneously in the right flank under isoflurane anesthesia. Tumor volumes were calculated using the equation shown below (tumor volume = major axis × minor axis^2^ × 0.52) from day 6 to 21.

For therapeutic antitumor experiments, C57BL6/N mice (6 weeks old) were anesthetized with isoflurane. E.G7-OVA cells (8 × 10^6^ cells/40 μL/mouse) were inoculated subcutaneously in the right flank. The tumor-bearing mice were intravenously injected with EGFP mRNA or OVA mRNA-loaded A-11-LNPs, OVA mRNA-loaded MC3-LNPs or OVA mRNA-loaded RNA-LPX at doses of 0.03 mg mRNA/kg on days 8 and 11. Tumor volumes were calculated by the above method from day 6 to 21.

### 2.11. Toxicity Test

C57BL6/N mice (6 weeks old) were intravenously injected with OVA mRNA-loaded A-11-LNPs at two doses of 0.03 mg mRNA/kg on days 0 and 3. Blood was obtained 24 h after the last dose and processed into plasma using heparin. Alanine transferase (ALT), aspartate transferase (AST), total bilirubin (T-BIL), lactate dehydrogenase (LDH), blood urea nitrogen (BUN), and creatinine (CRE) levels in plasma were measured at Oriental yeast Co., Ltd. (Shiga, Japan).

### 2.12. Statistical Analyses

Statistical data obtained by the design of the experiment (DOE) were analyzed using the JMP 14 software (SAS, Cary, NC, USA). Statistical significance was defined as *p*-values less than 0.05. Two independent experiments were performed for the DOE. To identify significant factors for each physicochemical property of the LNPs in a relatively vast experimental design space, a 3^4^ × 2^2^ definitive screening design (DSD) was used for screening A. Effective design-based model selection for DSD or the forward stepwise regression method with Akaike’s information criterion and finite correction (c-AIC) was applied to each response. The forward stepwise regression method with c-AIC was applied only in cases where the number of both statistically significant main factors and interactions between 2 factors were less than 3. Responses with ranges of several digits (i.e., gene expression, cellular uptake) were converted to logarithms in order to ensure linearity. For screening B, a 2^4^ fractional factorial design (FFD) was used. A standard least squares linear regression model was applied to each response.

Results are expressed as the mean + standard deviation (SD) or mean ± SD of independent repeats. For comparisons between the means of two variables, we used unpaired Student’s *t*-tests. For comparisons between multiple groups, we used one-way analysis of variance (ANOVA) with the Tukey–Kramer posthoc tests. These statistical analyses were done using the JMP 14 software.

## 3. Results

### 3.1. Optimization Strategy for Splenic DCs

Microfluidic technology has made it possible to reproducibly produce relatively small and uniform LNPs and has recently been adopted as a major production method [42]. The rapid and reproducible mixing of two liquids (an alcohol solution of lipids and a buffer solution containing RNAs) is achieved by introducing them at a high flow rate into a microfluidic device equipped with a micromixer. While most RNA-loaded LNPs that are formed by microfluidic technology are less than 100 nm in size [43,44,45,46], we recently reported that the addition of a salt (e.g., NaCl) to the RNA-containing buffer significantly contributed to the formation of LNPs with sizes in excess of 100 nm based on the Derjaguin–Landau–Verwey–Overbeek (DLVO) theory [39]. The addition of a salt reduces electrostatic repulsion and promotes fusion between the initially produced liposome-like cationic particles, resulting in the formation of larger LNPs in a salt concentration-dependent manner. Both cellular uptake and the functional delivery of siRNA and mRNA in splenic DCs were significantly higher when larger LNPs were used compared to the smaller counterparts with a consistent lipid composition. This can be explained by the fact that macropinocytosis, which is a unique pathway characterized by the nonspecific internalization of large amounts of extracellular fluid, is constitutively active in immature DCs and, therefore, relatively larger-sized particles (e.g., 200 nm or higher) would be beneficial for targeting DCs [47,48]. Although the finding suggests that an increase in LNP size is a significant factor for the efficient delivery of RNAs to splenic DCs, the optimal ranges of the size and other factors, including lipid composition, have not yet been clarified.

In this study, we first conducted 2 steps of DOE for screening and optimizing both synthetic conditions (NaCl concentration) and the lipid composition of LNPs for the delivery of mRNA to splenic DCs. The experimental scheme is represented in Figure 1. The physicochemical properties and cellular uptake of the LNPs, which would be mediator variables, were also measured in an attempt to understand how these factors contribute to the functional delivery (final output). Nluc mRNA was used to quantify the transgene expression level. The long intracellular half-life of the Nluc protein (>6 h) attenuates the potential effects of protein degradation during the lengthy (at least several hours) experimental process from sacrifice to cell sorting and would be suitable for the measurement of enzymatic activity in the sorted cells. The bright signal from Nluc was also suitable for quantitatively detecting transgene expression levels in the limited number of sorted cells. DiO-labeled LNPs were used to assess cellular uptake. Three types of APCs, including DCs, macrophages (Mφ), and B cells, were used in the analysis.

Adjuvant effects associated with type I interferon (IFN) (IFN-I) stimulation have been suggested to lead to superior acquired immune responses, and pathways that activate IFN-I expression have been identified, including the TLRs, RLRs, and stimulator of interferon genes (STING) pathways [12]. Mouse TLR7, which recognizes single-stranded RNA (ssRNA) and activates the adaptor protein myeloid differentiation primary response 88 (MyD88), leads to the expression of a suite of inflammatory cytokines, including IFN-I [49]. It is generally thought that regular uridine-containing mRNA, which is an ssRNA, stimulates the innate immune system via TLR7. The cytoplasmic RNA sensor RIG-I binds to double-stranded RNA (dsRNA) and induces the expression of inflammatory cytokines such as IFNβ through the activation of the adapter protein mitochondrial antiviral signaling protein (MAVS) [50]. Trace amounts of dsRNA, a byproduct of the in vitro transcription of mRNA, contribute to IFNβ production via the RIG-I/MAVS pathway [51,52]. In this study, we used regular uridine-containing mRNAs, which were obtained from Trilink BioTechnologies, which are thought to induce IFN-I expression through stimulation of the innate immune system via the RIG-I/MAVS pathway and TLR7. IFN-I is known to drive a distinctive DC maturation program, including the continuous upregulation of MHC-II and antigen processing [53], and to be highly expressed by DCs (especially by plasmacytoid DCs that highly and constitutively express IFN response factor-7) [54]. Therefore, the efficient introduction of immunostimulatory mRNAs into DCs results in the efficient maturation of DCs and, for dendritic cells, the expression level of MHC class II (I-A/I-E), one of the maturation markers, was also quantified in the screening.

### 3.2. DOE-Based Optimization of LNPs

In the 1st DOE (screening A), 6 independent factors, including the molar percentage of CL (level: 40 to 60), PL (level: 10 to 40), PEG-lipid (level: 0.5 to 1.5), a type of CL (level: CL4H6 or CL7H6), PEG-lipid (level: PEG-DMG or PEG-DSG), and NaCl concentration (level: 0 to 400 mM), were systematically examined. Since TFR and FRR had only a limited impact on the size of LNPs in our previous study and have substantial issues (reduced reproducibility at lower levels and unintended dilution at higher levels, respectively), these parameters were fixed at 500 µL/min and 3, respectively. DSD was adopted in screening A to determine the significant contributing factor(s) for the responses (including physicochemical properties, cellular uptake, and gene expression) and to narrow down experimental space of the following 2nd DOE (screening B) (Table 1). The RNA-loaded LNPs were synthesized under 14 different formulation conditions (coded as A-1 to A-14) that were determined based on DSD. The diameter of the LNPs varied with a ζ-average from 88 to 754 nm (Table 1). The LNPs were typically uniform (PdI of 0.2 or less) except for A-4 and A-5. The encapsulation efficiency and ζ-potential of the LNPs were typically high (85% or over) and near neutral (within ± 5 mV), respectively, except for A-14 (72.4% encapsulation and −10.3 mV). The reproducibility of the synthesis of the LNPs between 2 technically independent experiments were confirmed (Appendix A). Statistical analysis revealed that an increase in NaCl concentration had the most significant effect on increasing the size of the LNP, an observation that is consistent with our previous study (Appendix A). A decrease in both %DOPE and %PEG also had a significant effect on the increase in size (Appendix A). This was also consistent with our previous study [39] and can be explained by the fact that the relatively bulky hydrophilic moieties of both DOPE and PEG-lipid resist the decrease in total surface area of the LNP with increasing particle size. Both CL4H6 and a higher %DOPE significantly contributed to a higher encapsulation efficiency (Appendix A). It is possible that an oxygen atom in the morpholino ring at the hydrophilic head of CL7H6 attracts electrons from the tertiary amine, thereby lowering the acid-dissociation constant and reducing the RNA encapsulation efficiency, especially under competitive conditions in the presence of NaCl. DOPE can facilitate encapsulation of RNA due to the fact that a primary amino group of DOPE, which is a proton donor, can form direct hydrogen bonds with phosphate groups of RNAs [30].

Cellular uptake and Nluc activity in 3 types of splenic APCs and whole spleens for the 14 types of LNPs are summarized in Appendix A. The reproducibility of Nluc activity in the whole spleen, cellular uptake and Nluc activity in splenic DCs were confirmed (Appendix A). Scatter plots revealed that the cellular uptake in the 3 types of splenic APCs was well correlated (Figure 2A,B), suggesting that factors and their levels examined in screening A have moderate effects on cell specificity between the 3 types of splenic APCs. On the other hand, the amount of cellular uptake differed significantly (1 to 2 orders of magnitude) among the LNPs, and the order was Mφ > DCs >> B cells. Scatter plots of Nluc expression versus cellular uptake in DCs and Mφ showed a trend toward higher activity in DCs (Figure 2C). An analysis of covariance was performed to reveal statistical differences in the Nluc expression level between DCs and Mφ without the effect of cellular uptake level, and the findings revealed that Nluc expression in the DCs was significantly higher than that in Mφ, which is consistent with observations obtained in a previous report [2]. The slope of Nluc activity to cellular uptake was obviously higher than 1 for DCs, suggesting that more LNPs are taken up per cell or LNPs with characteristics that allow them to be more easily taken up lead to higher transgene expression in DCs. The maximum Nluc expression in B cells was less than 100 RLU/3000 cells, which was over 100-fold lower than that in both DCs and Mφ (Appendix A). Scatter plots of Nluc expression in the whole spleen vs. splenic DCs showed a moderate correlation (R^2^ = 0.3919) (Figure 2D). Although the transgene expression level in the whole spleen has been extensively measured in many studies on mRNA vaccine-oriented formulation development due to its simplicity [55,56,57], the data indicate that transgene expression level in the whole spleen would not be a good indicator of the corresponding process in DCs, even if the transgene expression level in DCs was high. This can be attributed to the fact that the population of splenic DCs is only ~1% of all splenocytes. Therefore, much lower Nluc signals derived from B cells (~50% of splenocytes) and other cell types that were not examined in this study would account for a non-negligible proportion.

Concerning Nluc activity in splenic DCs, a well-fitted regression model was obtained by the effective design-based model selection for DSD process (Figure 3A). The Nluc expression (RLU/3000 cells) was converted to logarithms before regression because the range of the Nluc activity spans 3 orders of magnitude. A total of 5 out of all 6 factors that were examined in screening A were found to significantly contribute to Nluc expression in splenic DCs (Figure 3B). CL showed the highest impact on the Nluc expression level, and the use of CL4H6 was preferable to CL7H6. A higher %CL was also important in terms of improving the Nluc expression level. In addition, lower %PEG and higher NaCl concentration were also significant in improving the Nluc activity. The cellular uptake and Nluc expression in splenic DCs were plotted against ζ-average because the two factors are also significant determinants of LNP size. The plot clearly indicated that cellular uptake sharply increases by the ζ-average for sizes of up to ~200 nm (Figure 3C), but the upward trend was less pronounced in the range above 200 nm. On the other hand, the cellular uptake of extremely large LNPs (>700 nm) by DCs tended to be decreased. The plot for Nluc expression also showed a similar trend to cellular uptake (Figure 3D). Particles that are too large have the potential to induce toxicity and cause the obstruction of pulmonary microvessels. Therefore, an optimal LNP diameter for splenic DCs would be in the range of 200 to 500 nm. These observations suggest that the ζ-average is an intermediate factor that links the examined factors and cellular uptake or transgene expression level. To confirm the validity of the above regression model, the worst 5 and the top 5 LNPs were selected from 14 LNPs that were tested in screening A, and the occupancy of each level in each factor was then visualized. Nluc activity, cellular uptake, and ζ-average between the 2 groups were significantly different (Figure 3E–G). For %PEG, NaCl concentration, and CL, the levels of occupancy were clearly reversed between the two groups (Figure 3H–J). For example, 80% CL7H6 and 80% CL4H6 were occupied in the worst 5 and the top 5, respectively, for CL (Figure 3J). These results are consistent with the regression model. For the remaining factors, including %CL, %PL, and PEG, differences between the 2 groups were not clear, while both %CL and PEG were significant in the regression model (Figure 3K–M). Given the above findings, PEG was again examined in the next DOE. The level of %CL was increased from 40–60 to 50–60%. The level of %PL was narrowed down to 10–20% to reduce its impact on the size of LNPs. The level of %PEG was narrowed down from 0.5–1.5 to 0.75–1.5% to avoid the production of extremely large LNPs. CL was fixed with CL4H6. NaCl concentration was determined to be 300 mM for producing LNPs with an appropriate size based on simulations using the regression model.

In screening B, 8 different LNPs were synthesized based on the FFD (Appendix A) and evaluated in vivo (Appendix A). The diameters of the LNPs were in the range of approximately 110 to 420 nm with a narrow distribution (PdI < 0.15) and high encapsulation (>80%). In vivo screening indicated positive correlations between LNP size and cellular uptake or transgene expression in splenic DCs (R^2^ = 0.8186 or 0.9173, respectively) (Appendix A), a trend similar to that observed in screening A. Substantial correlations were observed when data from screening A and B were combined (Appendix A). On the other hand, correlations between LNP size and transgene expression in the whole spleen were clearly lower (R^2^ = 0.1148) (Appendix A). The correlation between transgene activity in splenic DCs and the whole spleen was also low when data from screening A and B were combined (R^2^ = 0.3582) (Appendix A). These results suggest that transgene activity in the whole spleen is not a good indicator of the same process in splenic DCs, and that evaluation at the cell type level is essential for selecting the optimal formulation.

Based on the screenings, the A-11-LNPs showed the highest transgene activity in splenic DCs (Appendix A and Appendix A). The A-11-LNPs showed the most enhanced I-A/I-E expression in splenic DCs among all of the LNPs that were tested (Appendix A). The size and RNA encapsulation of the A-11-LNPs remained constant at 4 °C for at least 24 days after their preparation, indicating a stable formulation (Appendix A). Considering the above data, we conclude that the A-11-LNPs were the optimal formulation for targeting splenic DCs as mRNA vaccines.

### 3.3. Comparison with Clinically Relevant Formulations

We next tested the A-11-LNPs against two clinically relevant RNA-loaded LNP formulations to determine their potential as an mRNA vaccine formulation. The first was the MC3-LNP, also known as Onpattro, a first-ever siRNA therapeutic developed by Alnylam pharmaceuticals for the treatment of transthyretin (TTR) amyloidosis [58,59]. The MC3-LNP was developed as a siRNA formulation targeting liver tissue. The MC3-LNP is considered to be the gold standard LNP formulation and is also widely used as a formulation for mRNA delivery [17,57,60,61]. For these reasons, it was used as a comparative formulation in this study. The second is RNA-LPX, which was reported by Kranz, et al. as a formulation that shows excellent cancer vaccine efficacy by selectively introducing mRNA into splenic DCs [2]. We selected this formulation because it is oriented for the same application as the A-11-LNP, and, because of this, we considered it to be the most suitable control. The MC3-LNP was 59 nm in diameter and showed a 97% mRNA encapsulation. The RNA-LPX was 319 nm in diameter and negatively charged (ζ-potential of −17 mV), consistent with previous reports [2]. The transgene expression level in secondary lymphoid tissues, the spleen and inguinal LN, was next evaluated using Nluc mRNA. The A-11-LNPs showed significantly higher transgene expression levels in both tissues over the two control formulations (Figure 4A). The RNA-LPX showed a superior transgene expression in the spleen compared to MC3-LNPs, which would be reasonable because of the fact that the RNA-LPX was reported as a spleen-selective mRNA delivery system. However, and interestingly, the MC3-LNPs showed a much higher transgene expression in inguinal LN over the RNA-LPX. The transgene expression level in splenic DCs was then measured using EGFP mRNA. The A-11-LNPs achieved approximately 9% of EGFP^+^ DCs, which was significantly higher than the two control formulations, indicating the superior transgene expression potency of the A-11-LNPs (Figure 4B). The highest I-A/I-E expression in splenic DCs was also observed for the A-11-LNPs (Figure 4C). These results suggest that the A-11-LNP would have a superior mRNA vaccine effect.

Motivated by the above results, a prophylactic antitumor experiment in E.G7-OVA tumor-bearing mouse model was conducted. Two sequential administrations of OVA mRNA-loaded A-11-LNPs completely rejected the tumor establishment at all doses tested (Figure 5A). We next examined the therapeutic antitumor effect of OVA mRNA-loaded A-11-LNPs in the same tumor-bearing mouse model. Since a clear shrinkage of tumor tissues was observed at doses of 0.015 mg mRNA/kg or higher in the dose-dependent study (Appendix A), the dose in subsequent experiments was set at 0.03 mg mRNA/kg. A comparative study of the therapeutic antitumor effects of the A-11-LNP and two control formulations (MC3-LNP and RNA-LPX) showed that only the A-11-LNP exhibited significant antitumor activity (Figure 5B). Increased serum IFNβ levels were observed only after the administration of the A-11-LNPs (Appendix A). The A-11-LNPs induced the maturation of splenic DCs, which resulted in the upregulation of the activation markers CD40, CD80, and CD86 (Figure 5C–E). The A-11-LNPs also activated splenic B, T, and natural killer cells, which upregulated the activation marker CD69 (Figure 5F–H). No antitumor activity was observed in the case of the EGFP mRNA-loaded A-11-LNP (Figure 5B), indicating that the antitumor effect depends on the development of an OVA antigen-specific immune response. In addition, B-8-LNPs, which showed significantly lower activity (2.1%) in splenic DCs but exhibited comparable activity (73%) in the whole spleen compared to the A-11-LNPs (Appendix A and Appendix A), showed no significant antitumor effect (Figure 5B), suggesting that mRNA was efficiently delivered to DCs rather than to the whole spleen, thus leading to antitumor activity. The limited induction of I-A/I-E expression in splenic DCs by the B-8-LNPs also supports this suggestion (Appendix A). Although clear antitumor effects were observed for two doses (0.03 mg mRNA/kg × 2) of the A-11-LNPs, the tumors tended to re-grow after day18 without having completely disappeared (Figure 5B). Based on this observation, the number of doses and dosages were increased (0.125 to 0.5 mg mRNA/kg × 3) to examine the therapeutic antitumor effect. Although there was one individual that experienced a complete response at doses of 0.25 and 0.5 mg mRNA/kg, the results showed a limited increase in the overall antitumor effect (Appendix A). This suggests that the immunosuppressive pathways are enhanced in association with the induction of antitumor immunity by the A-11-LNPs. In fact, E.G7-OVA cells express programmed cell death ligand-1 (PD-L1) [62], one of the major immune checkpoint molecules, and the combination of an adjuvant and an anti-PD-L1 antibody, an immune checkpoint inhibitor (ICI), appears to promote E.G7-OVA tumor shrinkage compared to the use of the adjuvant alone [63,64]. Therefore, the combination of the A-11-LNPs and ICIs would be a reasonable strategy for enhancing an antitumor effect in this tumor model. In addition, when the same tumor cells were re-challenged at 68 days after the initial tumor transplantation in the two individuals that completely responded, they completely rejected tumor engraftment (Appendix A), suggesting that treatment with the A-11-LNPs induced the formation of memory cells, which was also observed for the RNA-LPX in the previous study [2].

Finally, a hematological test was performed after two doses of the OVA mRNA-loaded LNPs at 0.03 mg mRNA/kg had been administered, and no significant alteration in any of the serum chemistry parameters tested was observed, suggesting that the A-11-LNPs are well tolerated under therapeutically relevant doses (Figure 6).

## 4. Conclusions

In the present study, the DOE-based in vivo optimization of mRNA-loaded LNPs for mRNA vaccines was studied. The screening included a wide range of particle sizes that were controlled by the addition of NaCl in addition to the lipid type used and its composition. The results showed a clear correlation between particle size and uptake and gene expression activity in splenic DCs and indicated that a size range from 200 to 500 nm is appropriate for use in conjunction with splenic DCs. It was also found that transgene expression activity in the whole spleen, which can be easily evaluated, was not a valid predictor of transgene expression activity in splenic DCs and potency as mRNA vaccines. The A-11-LNP, which was found to have the optimal formulation, induced better transgene expression activity and maturation in DCs compared to two clinically relevant LNP formulations and induced clear therapeutic antitumor effects in an E.G7-OVA tumor model. The findings reported herein are expected to contribute to the development of future mRNA vaccines.

## 5. Patents

K. Sasaki, Y. Sato, and H. Harashima have filed intellectual property patents related to this publication.

## Figures and Tables

**Figure 1 pharmaceutics-14-01572-f001:**
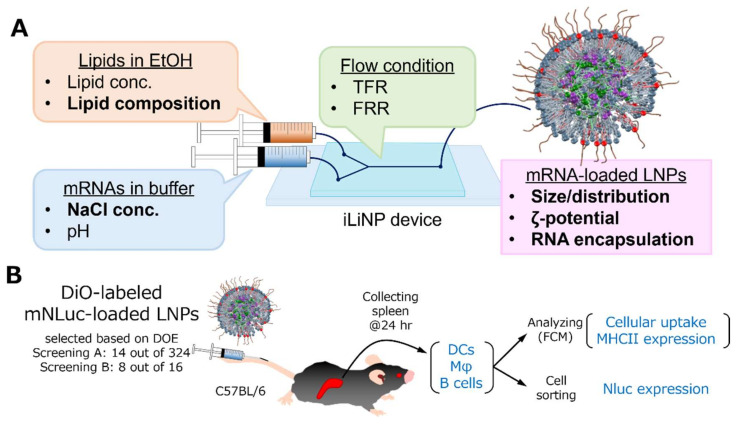
Schematic illustration of the experimental method for DOE. (**A**) Parameters and responses in manufacturing LNPs. Both parameters and responses examined in this study were expressed in bold. (**B**) Method for screening of LNPs in vivo. Cellular uptake, NHC class II expression, and Nluc expression in splenic DCs, Mφ, and B cells after intravenous injection of DiO-labeled mNLuc-loaded LNPs selected by DOE were measured to identify the optimal formulation.

**Figure 2 pharmaceutics-14-01572-f002:**
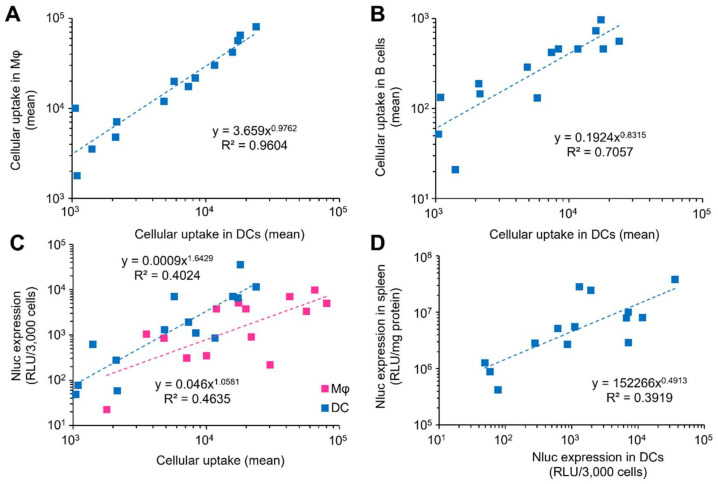
Correlations between outputs of in vivo experiments in screening A. (**A**) A dot-plot of cellular uptake in Mφ versus that in DCs. (**B**) A dot-plot of cellular uptake in B cells versus that in DCs. (**C**) A dot-plot of Nluc expression versus cellular uptake in both Mφ and DCs. (**D**) A dot-plot of Nluc expression in spleen versus in DCs.

**Figure 3 pharmaceutics-14-01572-f003:**
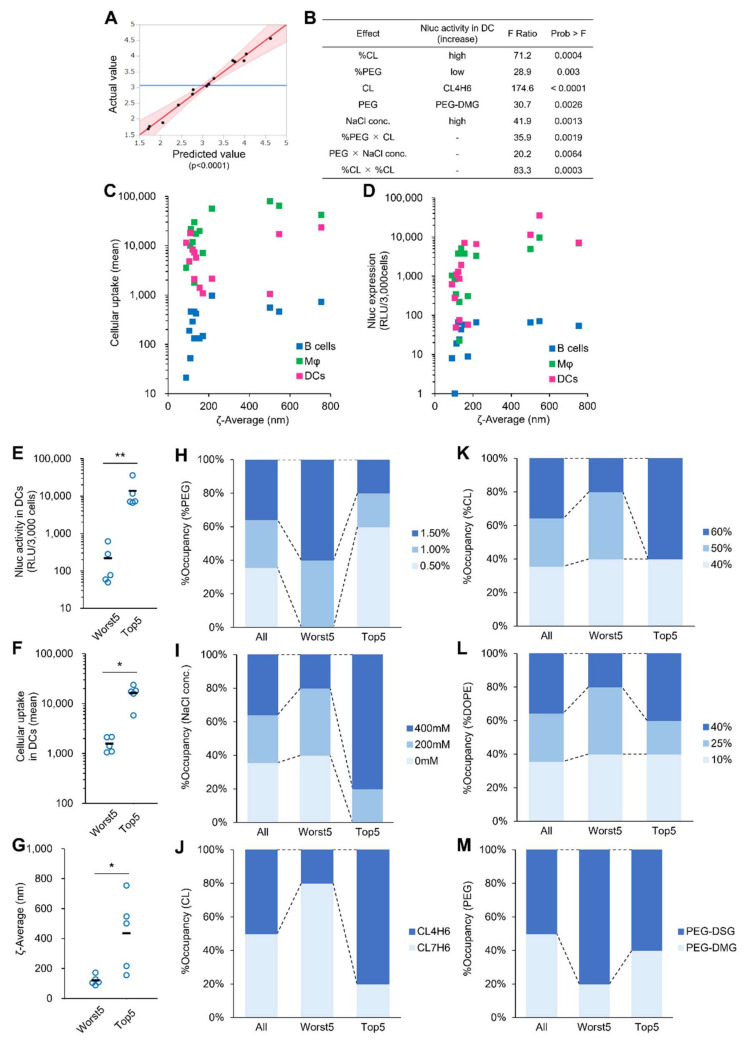
Analysis of significant factors for outputs in in vivo experiment in screening A. (**A**) A plot of predicted and actual values for Nluc expression in splenic DCs obtained by a regression model. (**B**) Statistically significant main factors and interactions for Nluc expression in splenic DCs. (**C**,**D**) Plots of cellular uptake (**C**) or Nluc expression (**D**) in splenic DCs versus ζ-average. Two independent experiments with one mouse in each preparation were performed. The plots are expressed as the average of the two experiments. (**E**–**G**) Comparison of Nluc expression (**E**), cellular uptake (**F**), and ζ-average (**G**) between top 5 and worst 5 LNPs on Nluc expression in splenic DCs. *n* = 5, * *p* < 0.05, ** *p* < 0.01. (**H**–**M**) Occupancy of each level of %PEG (**H**), NaCl conc. (**I**), CL (**J**), %CL (**K**), %DOPE (**L**), and PEG (**M**).

**Figure 4 pharmaceutics-14-01572-f004:**
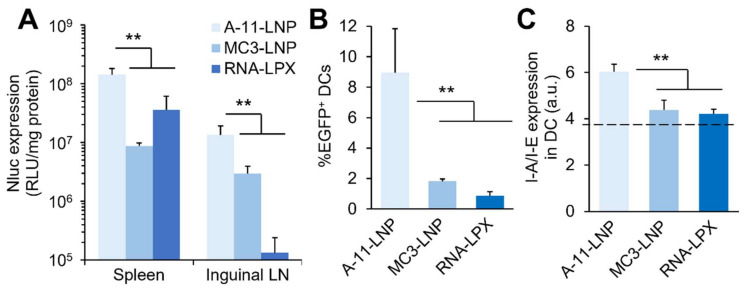
Comparison of the A-11-LNPs with clinically relevant formulations. (**A**) Nluc expression in spleen and inguinal LN 24 h after intravenous administration of Nluc-mRNA-loaded formulations at a dose of 0.5 mg mRNA/kg. (**B**) Percentage of EGFP^+^ splenic DCs 24 h after the intravenous administration of EGFP-mRNA-loaded formulations at a dose of 0.5 mg mRNA/kg. (**C**) I-A/I-E expression in splenic DCs 24 h after intravenous administration of EGFP-mRNA-loaded formulations at a dose of 0.5 mg mRNA/kg. *n* = 3. ** *p* < 0.01.

**Figure 5 pharmaceutics-14-01572-f005:**
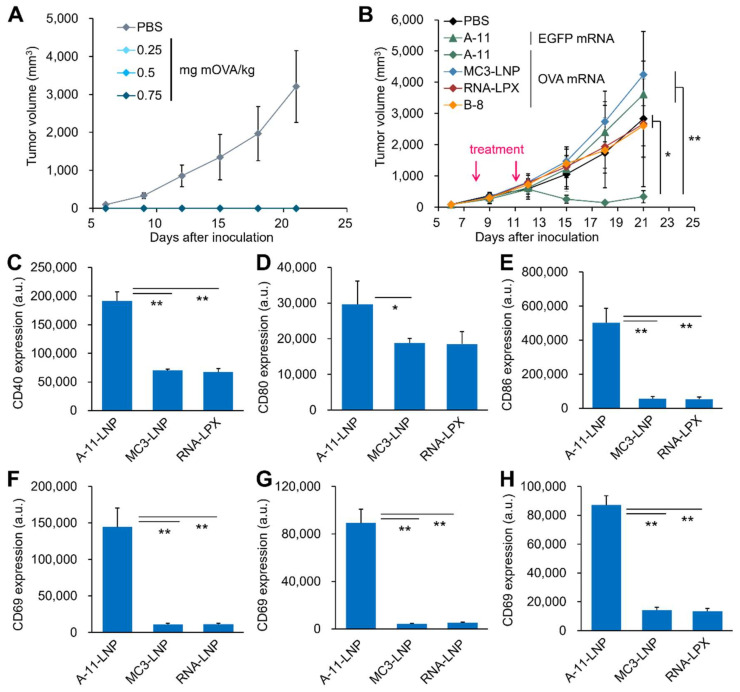
Prophylactic and therapeutic antitumor activity. (**A**) Prophylactic antitumor activity of the A-11-LNPs in E.G7-OVA tumor-bearing mice. The OVA mRNA-loaded A-11-LNPs were intravenously injected at the indicated doses twice on 14 and 7 days before tumor inoculation. *n* = 3–4. (**B**) The therapeutic antitumor activity of the A-11-LNPs, MC3-LNP, RNA-LPX, and B-8-LNPs. E.G7-OVA tumor-bearing mice were intravenously injected with OVA mRNA-loaded formulations at two doses of 0.03 mg mRNA/kg on day 8 and 11. *n* = 5. * *p* < 0.05, ** *p* < 0.01. (**C**–**E**) Expression of activation markers CD40 (**C**), CD80 (**D**), and CD86 (**E**) in splenic DCs 24 h after an intravenous injection of OVA mRNA-loaded formulations at a dose of 0.03 mg mRNA/kg. *n* = 3. * *p* < 0.05, ** *p* < 0.01. (**F**–**H**) Expression of an activation marker CD69 in splenic B cells (**F**), T cells (**G**), and NK cells (**H**) 24 h after an intravenous injection of OVA mRNA-loaded formulations at a dose of 0.03 mg mRNA/kg. *n* = 3. ** *p* < 0.01.

**Figure 6 pharmaceutics-14-01572-f006:**
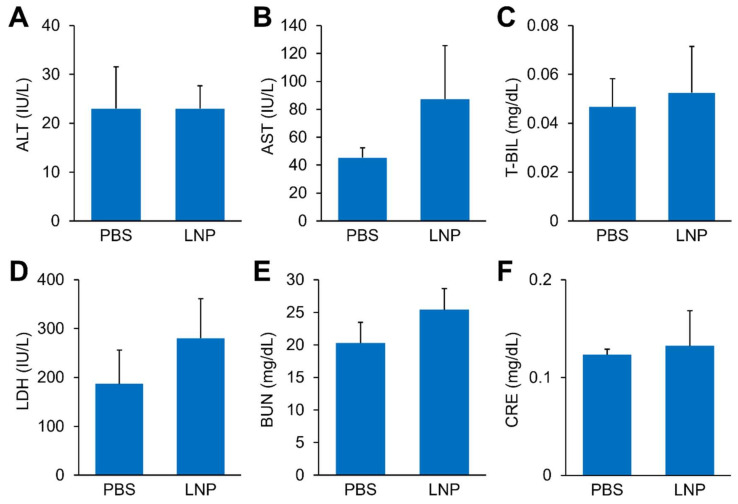
Safety of the A-11-LNPs. Serum chemistry parameters, ALT (**A**), AST (**B**), T-BIL (**C**), LDH (**D**), BUN (**E**), and CRE (**F**) were measured 24 h after the last injection of OVA mRNA-loaded A-11-LNPs at two doses of 0.03 mg mRNA/kg. *n* = 3.

**Table 1 pharmaceutics-14-01572-t001:** Physicochemical properties of the LNPs examined in screening A.

Entry	%CL	%DOPE	%PEG	NaCl Conc. (mM)	CL	PEG	ζ-Average (nm)	PdI	ζ-Potential (mV)	%RNA Encapsulation
A-1	40	40	0.5	0	CL7H6	PEG-DMG	112	0.20	−3.2	90.3
A-2	50	10	0.5	0	CL4H6	PEG-DMG	121	0.19	−2.3	99.2
A-3	60	10	1	0	CL7H6	PEG-DSG	104	0.15	−2.3	91.0
A-4	60	40	1.5	0	CL7H6	PEG-DMG	128	0.41	−0.6	89.8
A-5	40	25	1.5	0	CL4H6	PEG-DSG	88	0.30	−1.5	99.2
A-6	60	40	0.5	200	CL4H6	PEG-DSG	155	0.13	1.9	98.8
A-7	50	25	1	200	CL4H6	PEG-DMG	138	0.12	−1.1	99.0
A-8	40	10	1.5	200	CL7H6	PEG-DMG	128	0.20	−2.6	85.0
A-9	50	25	1	200	CL7H6	PEG-DSG	171	0.15	−2.7	88.0
A-10	40	40	1	400	CL4H6	PEG-DMG	216	0.08	2.5	99.0
A-11	60	10	1.5	400	CL4H6	PEG-DSG	547	0.19	−1.4	89.2
A-12	50	40	1.5	400	CL7H6	PEG-DSG	108	0.07	−1.7	88.3
A-13	60	25	0.5	400	CL7H6	PEG-DMG	501	0.12	−1.2	91.2
A-14	40	10	0.5	400	CL4H6	PEG-DSG	754	0.17	−10.3	72.4

## Data Availability

Not applicable.

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
