# Peer review of "mRNA-Loaded Lipid Nanoparticles Targeting Dendritic Cells for Cancer Immunotherapy"

_pharmaceutics, 2022, doi:10.3390/pharmaceutics14081572_

Round 1

Reviewer 1 Report

mRNA-loaded lipid nanoparticles targeting dendritic cells for 2 cancer immunotherapy.

Manuscript ID: pharmaceutics-1832994 

The authors have explored the possibility of targeting mRNA LNP vaccine formulations  specifically to the dendritic cells by modifying the size of the LNP’s. Dendritic cells (DCs) are attractive antigen presenting cells to be targeted for vaccinations. Currently the LNP vaccine formulations in the market do not specifically target the DC’s. Targeting them to the DC is an highly applicable and interesting research. The authors reported that reported that it is possible to regulate the size of RNA-loaded lipid nanoparticles (LNPs) to over 200 nm the addition of salt during their formation when a microfluidic device is used and that larger LNPs delivered RNA more efficiently and in greater numbers to splenic DCs compared to the smaller counterparts. The results from the in vivo and the in vitro experiments suggested that The results showed a clear 17 correlation between particle size, uptake and gene expression activity in splenic DCs, and indicated 18 that a size range from 200 to 500 nm is appropriate for use for targeting splenic DCs.

This is an highly translatable research with various clinical implications. The study has been presented is an excellent manner. The results and the methods ae perfectly explained in detail. Similarly, the introduction and the conclusion section are also explained perfectly. Although the manuscript is perfectly drafted, the reviewer would like to add the following minor comment for further improving the quality of the manuscript.

Minor comments:

1. Introduction line 71-74, looks incomplete.

Author Response

Response to Reviewer 1 Comments

Point 1: Introduction line 71-74, looks incomplete.

Response 1: Thank you for the comment. We meant to provide a clear explanation that the RNA-LPXs can achieve a selective gene expression in spleen without ligand modification in this sentence. Therefore, we added “ligand-unmodified” for clear contrast with the above-mentioned ligand-based targeting strategy as follows:

Line 72-74

On the other hand, Kranz et al. reported on an RNA-lipoplex (LPX) that was prepared by mixing mRNA and ligand-unmodified liposomes composed of…

Reviewer 2 Report

Thanks for your submission. The paper was interesting to read

Best of luck

Detailed minor comments:

I have read the article again and as I said I really like the idea that the authors are going for the new trend of RNA and advanced formulations (as we call them during teaching)

Lipid carriers are not a new thing but using the mRNA to transfer is a new trend nowadays, the article was using it and managed to prove a good results out of it

Here is my thoughts on it and some of the advices which I am not sure why I did not mention before as I have it ticked on my original printed copy

Line 88 the abbreviated lipid has not been introduced before so I am not sure what it is. I appreciate there is a reference to the previous manuscript but I would appreciate telling me what it is here

In section 2.4 it will be easier to call it microfluidic system and define the rest of parameters in the experiment such as how did they evaporate the organic phase after generating the nanoparticles?

Section 2.5 the RNA complex was formed though they could have encapsulated it rather than made a complex, in any case how did they wash off the unchelated particles. Did they use filter, dialysis bags? Etc?

Line 148 says intravenously injected, I am not sure how did that happen as with small mice we tend to go intraperitoneal not intravenously as they are really small, just a confirmation that it was IV will do

Line 153 HBSS not been explained, there are few abbreviations that were not described in full throughout the manuscript

Might be a good idea as well when you do an animal test to do a control plus sample not just sample

Line 210 to my memory ALT is alanine aminotransferase not transaminase same as AST aspartate aminotransferase so probably that one need checking

Stats were performed using software which I appreciate but what was calculated? ANOVA or T test or what? More details about the software

Line 230 is finally saying it is microfluidic, this could have happened earlier

232 there could be a discussion about the difference in flow rate on size and other factors which probably could have taken place anyway within the experiment to make it better

255-269 felt like introduction rather than explaining their own results

293-294 authors are talking about encapsulation in here though I am not sure how did they achieve it by just normal incubation, no mixing or anything. I am really doubting it is encapsulation, to check with author

There is no mentioning in any part to the number of repetition or in this case the number of mices used per preparation, the results in figure 3 may vary in the interpretation way depends on the size of sample, in addition a better explanation for the graph in the text might be needed as I struggled to know the meaning of each part

Comparison in figure 4 was shown with other product in the market which kinda reduce the need for a control in their own experiment

Figure 5 I don’t understand why graph D shows no significant between A-11-LNP and RNA-LPX unless there is something not mentioned in here

Author Response

Response to Reviewer 2 Comments

Point 1: Line 88 the abbreviated lipid has not been introduced before so I am not sure what it is. I appreciate there is a reference to the previous manuscript but I would appreciate telling me what it is here

Response 1: Your comment sounds reasonable. We added the full name of each lipid.

Point 2: In section 2.4 it will be easier to call it microfluidic system and define the rest of parameters in the experiment such as how did they evaporate the organic phase after generating the nanoparticles?

Response 2: Dialysis was performed to remove ethanol and to adjust the pH to neutral. Reflected by your comment, we added this explanation to be clear.

Point 3: Section 2.5 the RNA complex was formed though they could have encapsulated it rather than made a complex, in any case how did they wash off the unchelated particles. Did they use filter, dialysis bags? Etc?

Response 3: RNA-LPXs are formed by simply mixing RNAs and cationic liposomes, and are used directly without a purification step. The process is consistent with that in original report and is not original in this manuscript. However, we added this point to be clear.

Point 4: Line 148 says intravenously injected, I am not sure how did that happen as with small mice we tend to go intraperitoneal not intravenously as they are really small, just a confirmation that it was IV will do

Response 4: Only intravenous injections were conducted in this study. In the LNP field, intraperitoneal injection is not a common injection route in cases of developing LNPs for systemic administration because the biodistribution pattern between the two injection routes is clearly different.

Point 5: Line 153 HBSS not been explained, there are few abbreviations that were not described in full throughout the manuscript

Response 5: According to the reviewer’s criticism, we added some additional abbreviations especially in section 2.

Point 6: Might be a good idea as well when you do an animal test to do a control plus sample not just sample

Response 6: We strongly agree with the importance of placing a control. On the other hand, we believe that the validity of the experimental system has been confirmed in past reports by comparison with clinically relevant formulations in Figure 4 and 5C-H.

Point 7: Line 210 to my memory ALT is alanine aminotransferase not transaminase same as AST aspartate aminotransferase so probably that one need checking

Response 7: We corrected these terms according to the reviewer’s comment.

Point 8: Stats were performed using software which I appreciate but what was calculated? ANOVA or T test or what? More details about the software

Response 8: As we mentioned in section 2.13, the unpaired Student’s t tests were performed for comparisons between the means of two variables. One-way analyses of variance (ANOVA) with the Tukey–Kramer post hoc tests were performed for comparisons between multiple groups. According to the reviewer’s comment, we added additional information concerning the software.

Point 9: Line 230 is finally saying it is microfluidic, this could have happened earlier

Response 9: According to the reviewer’s suggestion, we added information concerning the microfluidic device in the introduction part.

Point 10: 232 there could be a discussion about the difference in flow rate on size and other factors which probably could have taken place anyway within the experiment to make it better

Response 10: The impact of total flow rate and flow rate ratio was examined in our previous study (Okuda K et al., J Control Release, 348: 648-659, 2022). Although not considered in this study, we added additional comments on why we did not consider these factors in section 3.2, as they are important factors for size regulation.

Point 11: 255-269 felt like introduction rather than explaining their own results

Response 11: The explanation was a bit lengthy due to the multiple pathways involved, which may have given the impression of an "introduction. However, we believe this is a necessary explanation to permit the mRNAs used in this study to be properly understood and to understand the mechanism by which they induce an immune response.

Point 12: 293-294 authors are talking about encapsulation in here though I am not sure how did they achieve it by just normal incubation, no mixing or anything. I am really doubting it is encapsulation, to check with author

Response 12: In the production of RNA-loaded LNPs using microfluidic devices, the rapid mixing of the RNA solution (acidic buffer) and the lipid solution causes RNA to be encapsulated within the LNPs through complexation by electrostatic interaction between RNA and lipids and subsequent self-assembly of nanoparticles due to the increased polarity of the solution. This is a well-established technique that is widely used in this field, and the RNA encapsulation rate was measured by the well-established Ribogreen assay.

Point 13: There is no mentioning in any part to the number of repetition or in this case the number of mices used per preparation, the results in figure 3 may vary in the interpretation way depends on the size of sample, in addition a better explanation for the graph in the text might be needed as I struggled to know the meaning of each part

Response 13: Thank you for pointing this out. We performed two independent experiments with one mouse in each preparation and expressed the results as the average of the two experiments. Since the reproducibility between experiments has been confirmed (Figure S4), we see no problem in interpreting the results. We added an explanation to the Figure legend concerning this.

Point 14: Comparison in figure 4 was shown with other product in the market which kinda reduce the need for a control in their own experiment

Response 14: We agree with the reviewer.

Point 15: Figure 5 I don’t understand why graph D shows no significant between A-11-LNP and RNA-LPX unless there is something not mentioned in here

Response 15: While we expected to see a significant difference, perhaps due to the relatively larger error bar for RNA-LPX, this did not happen and we did not see a significant difference.